# Assessment of Risk of Bias in Osteosarcoma and Ewing's Sarcoma Randomized Controlled Trials: A Systematic Review

**Robert Koucheki** [1,2,*], **Aaron M. Gazendam** [3], **Jonathan R. Perera** [4,5], **Anthony Griffin** [5], **Peter Ferguson** [5,6], **Jay Wunder** [5,6] and **Kim Tsoi** [5,6]

1 Temerty Faculty of Medicine, University of Toronto, Toronto, ON M5S 1A8, Canada
2 Institute of Biomedical Engineering, University of Toronto, Toronto, ON M5S 3G9, Canada
3 Division of Orthopaedic Surgery, McMaster University, Hamilton, ON L8S 4L8, Canada; Aaron.gazendam@medportal.ca
4 Royal National Orthopaedic Hospital, NHS Trust, Brockley Hill, Stanmore, London HA7 4LP, UK; jonathan.perera@nhs.net
5 Department of Orthopaedic Surgery, Mount Sinai Hospital, Toronto, ON M5G 1X5, Canada; Anthony.Griffin@sinaihealth.ca (A.G.); Peter.Ferguson@sinaihealth.ca (P.F.); jay.wunder@sinaihealth.ca (J.W.); Kim.Tsoi@sinaihealth.ca (K.T.)
6 Division of Orthopaedic Surgery, University of Toronto, Toronto, ON M5T 1P5, Canada
* Correspondence: Robert.koucheki@mail.utoronto.ca

**Abstract:** Aim: The aim of this study was to systematically assess the risk of bias in osteosarcoma and Ewing's sarcoma (ES) randomized controlled trials (RCT) and to examine the relationships between bias and conflict of interest/industry sponsorship. Methods: An OVID-MEDLINE search was performed (1976–2019). Using the Cochrane Collaboration guidelines, two reviewers independently assessed the prevalence of risk of bias in different RCT design domains. The relationship between conflicts of interest and industry funding with the frequency of bias was examined. Results: 73 RCTs met inclusion criteria. Prevalence of low-risk bias domains was 47.3%, unclear-risk domains 47.8%, and 4.9% of the domains had a high-risk of bias. Domains with the highest risk of bias were blinding of participants/personnel and outcome assessors, followed by randomization and allocation concealment. Overtime, frequency of unclear-risk of bias domains decreased ($\chi^2 = 5.32$, $p = 0.02$), whilst low and high-risk domains increased ($\chi^2 = 8.13$, $p = 0.004$). Studies with conflicts of interest and industry sponsorships were 4.2 and 3.1 times more likely to have design domains with a high-risk of bias ($p < 0.05$). Conclusion: This study demonstrates that sources of potential bias are prevalent in both osteosarcoma and ES RCTs. Studies with financial conflicts of interest and industry sponsors were significantly more likely to have domains with a high-risk of bias. Improvements in reporting and adherence to proper methodology will reduce the risk of bias and improve the validity of the results of RCTs in osteosarcoma and ES.

**Keywords:** randomized controlled trial; risk of bias; osteosarcoma; Ewing's sarcoma

## 1. Introduction

Randomized controlled trials (RCTs) are the cornerstone of modern evidence-based medicine. In the 1970s, the medical scientific community entered a new era, when the Food and Drug Administration (FDA) required pharmaceutical companies to submit RCTs for new drug applications [1]. Since that time, RCTs have become the gold-standard for assessing the safety and efficacy of experimental therapies and interventions [1]. Bias refers to systematic errors leading to deviation of results that can cause over- or under-estimation of the true effect of an intervention. Given this, bias has the potential to undermine RCT findings and may limit the utility of the trial in clinical practice.

The Cochrane Collaborations tool for assessing risk of bias in RCTs is the most widely utilized and recognized tool used to critically appraise RCTs (Table 1) [2]. This tool assesses

the various methodological "domains" that are potential sources of bias in RCTs. Each domain is then rated based on its methodological quality as *low risk, high risk* or *unclear risk of bias* as per the guidelines set out by the Cochrane tool (Table 2). *Unclear risk* indicates "either a lack of information or uncertainty over the potential for bias" in a specific domain. According to the Preferred Reporting Items for Systematic Reviews and Meta-Analyses (PRISMA) guidelines, a risk of bias assessment of included trials must be performed as part of systematic reviews [3]. This tool has been utilized widely across the medical literature and can provide an understanding of the quality of RCTs at both the individual level and an appraisal of the quality of the literature within a particular field as a whole. Risk of bias assessments have been undertaken in both surgical and oncology trials [4,5]. However, to our knowledge, no studies have evaluated the presence of bias in RCTs focused on osteosarcoma and Ewing's Sarcoma (ES).

**Table 1.** Description and examples of the Cochrane Collaboration's seven domains [6,7].

| Domain | Type of Bias Addressed | Description | Example of *Low Risk* Characteristics | Example of *High Risk* Characteristics |
|---|---|---|---|---|
| Random Sequence Generation | Selection Bias | Addresses whether there were sufficient information describing the method used by the RCT to generate the allocation sequence. | • Random number tables <br> • Use of an electronic random number generator | • Sequence generation by date of birth <br> • Sequence generation by rule based on date of admission |
| Allocation Sequence Concealment | Selection Bias | Addresses whether there were sufficient information describing the method used to mask the allocation sequence. | • Central allocation <br> • Use of sealed envelopes | • Using an open random allocation |
| Blinding of Participants and Personnel | Performance Bias | Describes whether the participants and personnel were unaware of the interventions that the participants received. | • Clear statement of blinding/masking participants and personnel. | • No blinding/incomplete blinding |
| Blinding of Outcome Assessment | Detection Bias | Describes measures used to blind outcome assessors to interventions that the participants received. | • Clear statement of blinding/masking of outcome assessors | • No blinding of outcome assessment |
| Incomplete Outcome Data | Attrition Bias | Describes the completeness of outcome data for each major outcome. | • No missing outcome data | • Significant missing outcome data, which likely is related to true outcome |
| Selective Outcome Reporting | Reporting Bias | Describes reporting of all primary and secondary outcomes discussed within the introduction or methods section of the RCT. | • All of the study's pre-specified outcomes are reported | • Some of the study's pre-specified outcomes are missing |
| Other Sources of Bias | | State any important concerns about validity of the study not addressed elsewhere. | - | • Poor study design |

**Table 2.** Interpretation of Risk of Bias Ratings [6,7].

| Risk of Bias Rating | Interpretation |
| --- | --- |
| *Low Risk* | Interpreted as potential bias unlikely to affect the results. |
| *Unclear Risk* | Interpreted as potential bias that raises some concerns about the results. |
| *High Risk* | Interpreted as potential bias that seriously reduces confidence in the results. |

A source of bias not considered in the Cochrane Handbook is impact of industry funding and author conflicts of interest on methodologic quality and reported outcomes. RCTs are frequently supported by industry sponsors leading to potential conflicts of interest in these clinical studies [8]. It has been previously demonstrated that studies on the efficacy of drugs and devices, which are supported by manufacturing companies, report higher efficacy [9]. To our knowledge, the association between conflicts of interest and industry sponsorship with risk of bias in RCTs has not been previously assessed in orthopedic oncology studies.

RCTs for patients with primary bone tumors have unique challenges and a focused critical appraisal of this literature is warranted. Osteosarcoma and ES are rare entities, which has been shown to increase the bias, particularly in blinding and sample sizes [10,11]. The mainstay of definitive management of localized primary bone tumors includes surgical excision. Surgical trials pose many methodological challenges that are not always present in medical trials, which has the potential to introduce biases and reduce the validity of the results [12].

The primary objective of this study is to determine the prevalence of risk of bias in primary bone cancer RCTs. Secondarily, the impact of conflict of interest and industry sponsorship on risk of bias will be investigated. Finally, an evaluation of the change of the quality of RCTs over time will be undertaken.

## 2. Methods

A systematic review was performed on RCTs evaluating osteosarcoma and ES. This review adhered to the recommendations outlined in the PRIMSA and Cochrane Collaboration guidelines for the reporting of systematic reviews [3,6].

### 2.1. Eligibility Criteria

The inclusion and exclusion criteria were defined *a priori.* Inclusion criteria were as follows: (1) prospective study that had a parallel or cross-over longitudinal design, (2) studies examining a causal relationship between interventions and outcomes, (3) control or comparative group, and (4) presence of any number of included any cases of osteosarcoma and/or ES. Exclusion criteria included: (1) nonrandomized trials, (2) studies involving other diseases that could not be stratified by disease, (3) studies without online access (4) non-human trials, and (5) non-English studies.

### 2.2. Search Strategy

To systematically assess the risk of bias in published RCTs in osteosarcoma or ES, an Ovid MEDLINE (1946–2019) search was performed on 13 April 2020. Keywords included "Osteosarcoma" or "Ewing's Sarcoma". The search was limited by publications designated as "Randomized Controlled Trials".

### 2.3. Study Selection

The title and abstract of each article were screened and studies that did not meet the inclusion criteria were excluded. Full-text review was then performed for final assessment of study eligibility.

### 2.4. Data Extraction

Data extracted from the included studies was entered into a collaborative spreadsheet (Microsoft Excel V16.40). Study characteristics were recorded, including the publishing journal and impact factor, year of publication, conflicts of interests, and reported industry sponsorships. Trials were stratified by type: surgical vs. medical. Conflicts of interest were given one of three possible ratings: (1) No conflict of interest, when the study clearly stated that no conflict of interest was present, (2) Unclear conflict of interest, when there was no mention of the presence or absence of conflicting interests, (3) Conflict of interest, when mentioned. The same methodology was used for rating industry sponsorships.

Next, one reviewer (RK) assessed and appraised each study for its risk of bias using the *Cochrane Handbook for Systematic Reviews of Interventions* version 5.2 (2017)'s seven design domains. A senior author (JP) appraised the results to ensure accuracy and completeness. The domains are as follows: (1) Random sequence generation, (2) Allocation concealment, (3) Blinding of participants and personnel, (4) Blinding of outcome assessment, (5) Incomplete outcome data, (6) Selective outcome reporting, (7) Other potential threats to validity (Table 1). Each domain was designated with a *low*, *unclear*, or *high risk* of bias rating as per the Cochrane Handbook (Table 2) [7]. Disagreements were resolved by consensus.

### 2.5. Statistical Analysis

The presence, type, and rate of bias were summarized descriptively. The rate and type of bias present in each of the domains were also presented. Association between presence or absence of conflict of interests and industry sponsorships with the frequency of *high risk* domains was evaluated. The Fisher's exact test was used to measure the statistical significance in the analysis of contingency tables. An analysis of the risk of bias over time was performed to assess if bias has decreased over time. The frequency of *low risk*, *unclear*, and *high risk* of bias ratings were calculated for studies "prior to 1996", "1996–2000", "2001–2009", "2010–present" in correspondence with the *Consolidated Standards of Reporting Trials* (CONSORT) statement and its subsequent revisions published in 2001 and 2010 [13]. Pearson Chi-squared analyses were used to evaluate non-random statistical changes in the distribution of bias over time. Statistical significance was set at *p*-value < 0.05. GraphPad Prism V8.4.3 and Microsoft Excel V16.46 was used to prepare figures and perform the statistical analysis.

## 3. Results

The results of the search strategy are outlined in Figure 1. Overall, 164 studies were returned by the OVID MEDLINE search. After title and abstract screening, 91 studies did not meet inclusion criteria and were excluded. 73 studies met all inclusion criteria and moved forward for full text analysis and appraisal (Appendix A Table A1). Of the included studies 24 studies were published between 2009–2019, 22 studies were published between 2001–2009, 9 studies were published between 1996–2000, and 18 studies were published prior to the year 1996. The earliest study was published in 1976.

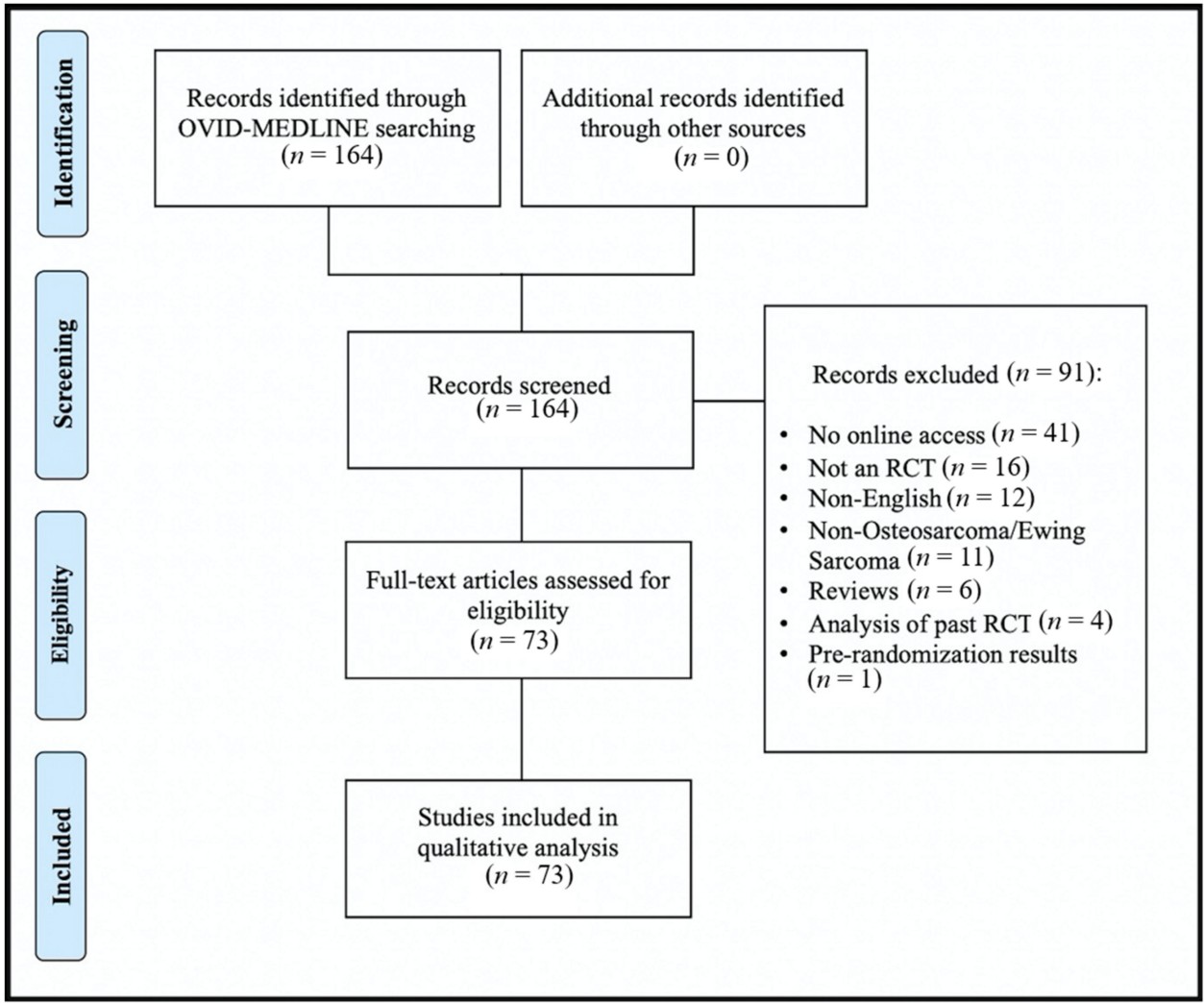

**Figure 1.** Flow diagram of included/excluded studies.

Among the included studies, 94.5% (69/73) of trials were medical and 5.5% (4/73) were surgical trials. (Figure 2A). With 13 RCTs, the "*Journal of Clinical Oncology*" had the most included trials, followed by "*International Journal of Radiation Oncology, Biology, Physics*" with 5 RCTs, and "*Cancer*" with 4 RCTs. Other journals with frequent osteosarcoma and ES trials are displayed in Table 3. Stratification of overall risk of bias stratified by type of sarcoma (osteosarcoma vs. Ewing sarcoma), type of intervention (medical vs. surgical), and presence or absence of metastasis is displayed in Appendix A Table A2.

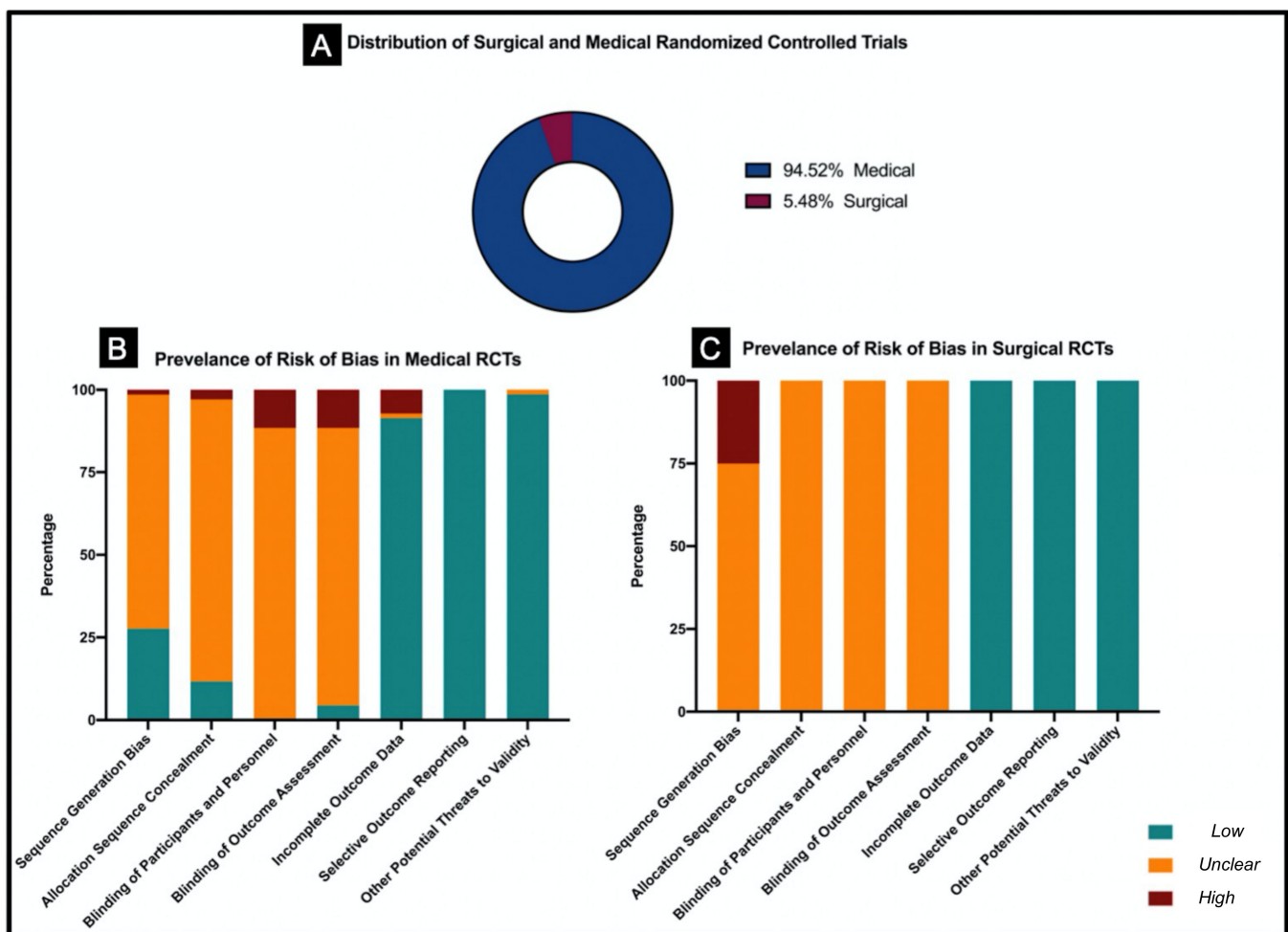

**Figure 2.** Comparison of risk of bias in medical and surgical RCTs regarding osteosarcoma and Ewing's sarcoma (**A**). Distribution of medical and surgical RCTs. (**B**). Prevalence of risk of bias in medical RCTs. (**C**). Prevalence of risk of bias in surgical RCTs.

**Table 3.** Most common journals for osteosarcoma and Ewing's sarcoma RCTs, with 2019 impact factors and percentage of unclear and high-risk domains.

| Journal | Number of RCTs | Impact in 2019 | Percentage of Domain with Unclear Risk | Percentage of Domain with High Risk |
|---|---|---|---|---|
| Journal of Clinical Oncology | 13 | 32.956 | 43.96% (40/91) | 6.59% (6/91) |
| International Journal of Radiation Oncology, Biology, Physics | 5 | 5.859 | 54.29% (19/35) | 2.86% (1/35) |
| Cancer | 4 | 5.742 | 57.14% (16/28) | 0.00% (0/28) |
| The Lancet Oncology | 3 | 33.752 | 9.52% (2/21) | 19.05% (4/21) |
| European Journal of Cancer | 3 | 7.275 | 47.62% (10/21) | 0.00% (0/21) |
| Annals of Oncology | 3 | 18.274 | 47.62% (10/21) | 9.52% (2/21) |
| The New England Journal of Medicine | 2 | 74.699 | 50.00% (7/14) | 0.00% (0/14) |
| Pediatric Blood & Cancer | 2 | 2.355 | 57.14% (8/14) | 7.14% (1/14) |
| Clinical Orthopaedics and Related Research | 2 | 4.091 | 28.57% (4/14) | 21.43% (3/14) |
| British journal of cancer | 2 | 5.791 | 57.14% (8/14) | 0.00% (0/14) |

Among the included RCTs, 47.8% of domains had an *unclear risk* of bias, 47.3% had a *low risk* of bias, and 4.9% had a *high risk* of bias. Of all the studies appraised, 93.15% (68/73) had at least one *unclear risk* domain and 20.5% (15/73) had at least one domain with a *high risk* of bias (Appendix A Table A3). The prevalence of risks of bias in different design domains are depicted in Figure 3. Across all of the appraised RCTs, the domain "selective outcome reporting" was found to have the lowest risk of bias. In contrast, for the blinding of participants and personnel domain no study adequately met *low risk* criteria and were either *unclear risk* (89.04%) or *high risk* (10.96%). Similarly, blinding of the outcome assessment had high rates of bias.

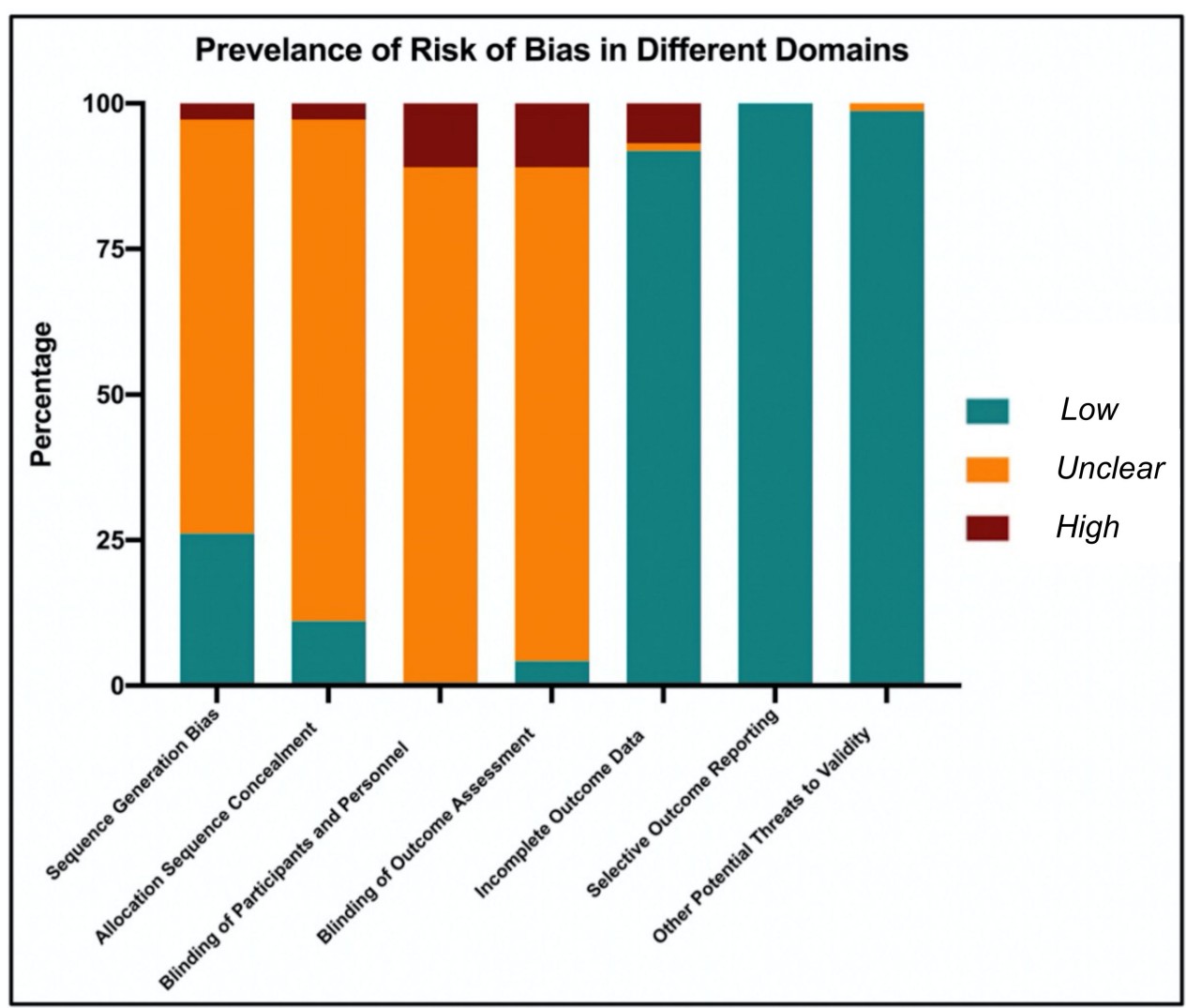

| | Sequence Generation Bias | Allocation Sequence Concealment | Blinding of Participants and Personnel | Blinding of Outcome Assessment | Incomplete Outcome Data | Selective Outcome Reporting | Other Potential Threats to Validity |
|---|---|---|---|---|---|---|---|
| *Low risk of bias* | 26.03% | 10.96% | 0% | 4.11% | 91.78% | 100% | 98.63% |
| *Unclear risk of bias* | 71.23% | 86.3% | 89.04% | 84.93% | 1.37% | 0% | 1.37% |
| *High risk of bias* | 2.74% | 2.74% | 10.96% | 10.96% | 6.85% | 0% | 0% |

**Figure 3.** Prevalence of risk of bias in seven different domains among all RCTs published in osteosarcoma and Ewing's Sarcoma.

For medical RCTs, 47.4% of the domains had an *unclear risk* of bias, 47.6% had a *low risk* of bias, and 5.0% of domains had a *high risk* of bias. For surgical RCTs, 53.6% of domains had an *unclear risk* of bias, 42.9% of domains had a *low risk* of bias, and 3.6% had a *high risk* of bias. Comparison of prevalence of risk of bias in surgical and medical RCTs are shown Figure 2B,C. Compared to medical RCTs, no significant difference in the risk of bias was found in surgical RCTs.

Conflicts of interest were present in 8.22% of studies, while 64.38% of studies did not have any reported conflict of interest. In the remainder of studies, it was not clearly stated whether a conflict of interest was present or not (Figure 4). Industry sponsorship was present in 8.22% of RCTs, while 38.36% of studies had no sponsorship or sponsors were not from the industry. Compared to RCTs with no conflicts of interest, studies that had a conflict of interest were 4.16 times more likely to have a *high risk* domain (Range: [1.56 to 11.47], $p = 0.01$). Additionally, compared to studies without an industry sponsor, studies with a sponsor were 3.06 times more likely to have a *high risk* domain ([1.15 to 8.34], $p = 0.03$).

The frequency of the risk of bias over time is displayed in Figure 5. Frequency of unclear risk domains decreased, while both *low risk* and *high risk* domains increased over time. Chi-square test for trends showed a decrease in *unclear risk* domain ($\chi^2 = 5.32$, $p = 0.02$) and an increase in *high risk* domains ($\chi^2 = 8.13$, $p = 0.004$).

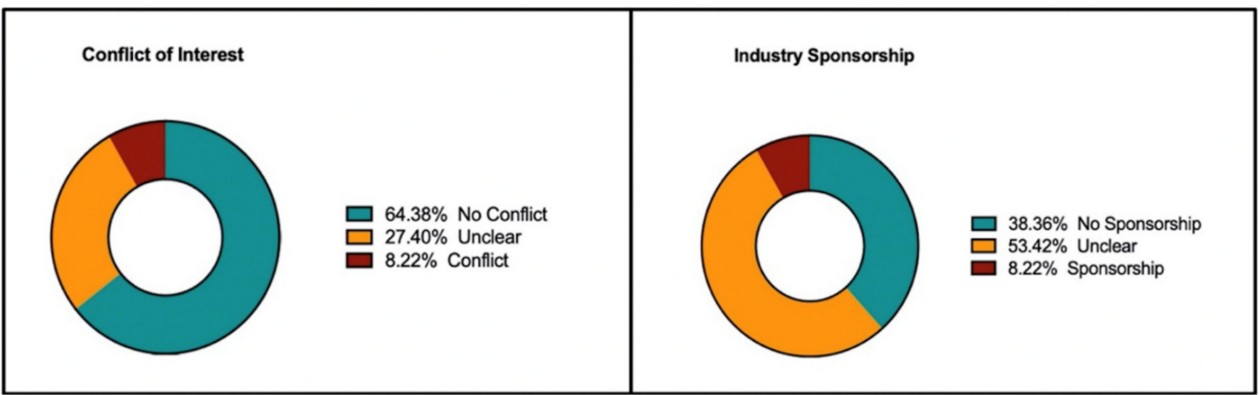

**Figure 4.** Prevalence of conflict of interest and industry sponsorship.

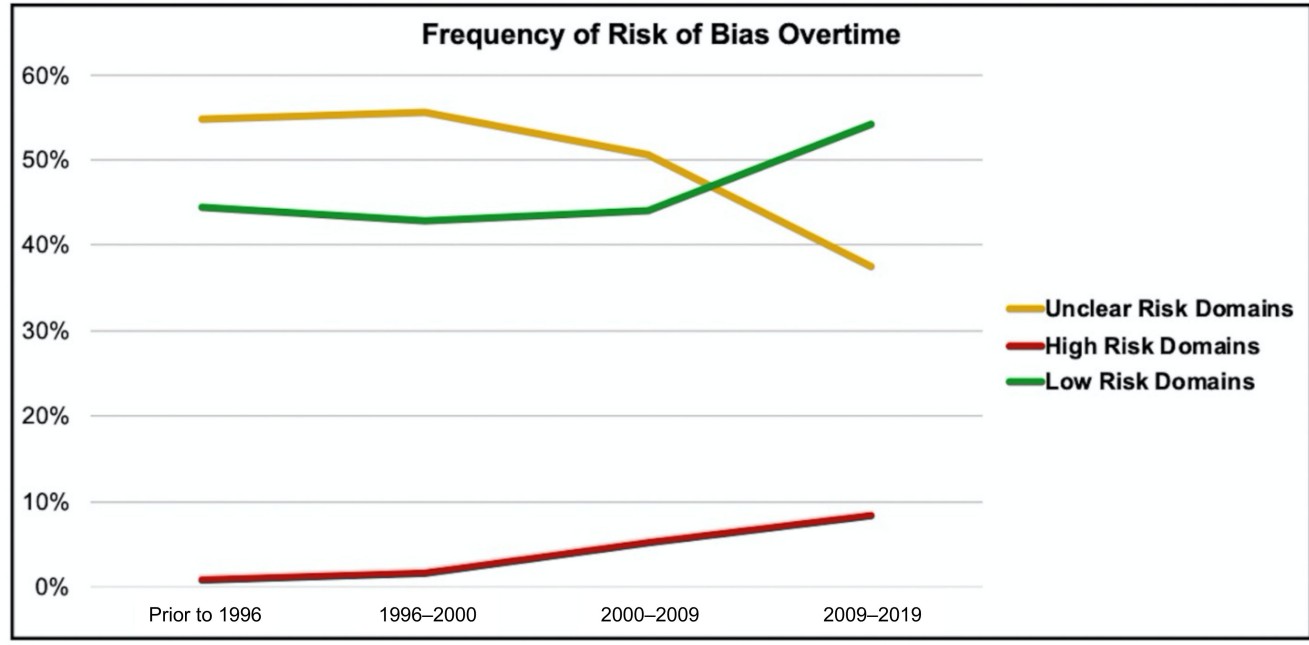

**Figure 5.** Frequency of risk of bias overtime.

## 4. Discussion

In this systematic review of 73 RCTs, 47.8% of bias domains had an *unclear risk* of bias and 4.9% of domains had a *high risk* of bias. The domains with the most risk of bias were blinding of participants and personnel and blinding of outcome assessors, followed by randomization and allocation concealment. Studies were significantly more likely to have a *high risk* of bias domain if author conflicts of interest or industry funding were present. These findings are in line with other areas of the literature including orthopedic surgery, ophthalmology, and plastic surgery [14–16].

Randomization is the cornerstone of RCTs and ensures comparative groups have similar known and unknown prognostic factors so causality can be established [17]. It is critical for study designers to properly randomize studies and to conceal the sequence allocation to maintain the internal validity of the study results. In 71.2% of the appraised studies an *unclear risk* of bias was present for randomization and sequence generation. Another frequent source of risk of bias was from allocation sequence concealment. Allocation is related to, but different than randomization and refers to the practice of keeping researchers unaware of the sequence of randomization until the moment of assignment. When the allocation process is not concealed, investigators may (knowingly or unknowingly) choose to enroll only certain eligible patients in a trial [18]. This can lead to biased estimates of the treatment/intervention effect [19,20]. Allocation concealment is always possible in RCTs and should be explicitly stated to reduce selection bias [21].

Blinding is an important methodological feature in the design of RCTs and attempts to reduce performance and detection bias. In the current review, 89% of studies were found to have *unclear risk* of bias and the remaining 11% were found to be of *high risk* of bias with respect to blinding of both participants and personnel. This high rate of bias is concerning and likely multifactorial. Firstly, trialists often fail to adequately describe which individuals in the study are blinded [12]. Using the term "blinded" or "double-blinded" is vague and fails to tell the reader which individuals involved in the study were blinded [22]. Secondly, 5.5% of the trials included surgical interventions, making blinding inherently more difficult for a number of reasons. Surgical interventions are often more difficult to blind than drugs trials, as placebos would require sham surgery [12]. Similarly, if the trial is comparing intraoperative methods, blinding of the surgeon becomes nearly impossible. However, despite these challenges, researchers should aim to blind as many of the involved individuals as possible and explicitly state how blinding is achieved in the methodology [23]. Although blinding of the treating surgeon and patient may be challenging or impossible, the blinding of outcome assessors and data analysts is usually possible and has been demonstrated successful in recent trials in the field [24,25]. Given that a lack of blinding is associated with more significant trial outcomes, it remains an important area to assess and scrutinize [23].

Financial conflicts of interest and industry sponsors are frequently found in clinical research, and are more common in oncology trials [8]. A review of both medical and surgical RCTs demonstrated that industry funded trials had significantly higher rates of positive trials in favor of the new industry product [26]. However, it is important to note that in the setting of rare tumors, alternative statistical methods are often utilized and these results may not be generalizable to this population. Adding further uncertainty, RCTs inconsistently report the presence or absence of conflicting interests and industry sponsors [27]. Our review is in line with these findings as studies with conflicts of interests were over four times more likely to have a *high risk* domain and if an industry sponsor was present the studies were over three times more likely to have a *high risk* domain ($p < 0.05$). Additionally, a limitation to the quality assessment of the trials was that 27.4% of studies did not discuss the presence or absence of conflicts of interests and over half of the trials did not report the source of their funding. Improved approaches are required for the identification and verification of conflicts of interest and sources of sponsorship [8].

The frequency of domains with *unclear bias* decreased over time. This is likely partially attributed to the widespread adoption of the CONSORT statement by both journals and researchers [28]. The CONSORT statement was first introduced in 1996 and trials published prior to this should be viewed through a different lens given the lack of reporting standards prior to this. This is encouraging and indicates improved reporting given that a large proportion of the *unclear bias* category is due to lack of reporting and not frank concerns with study methodology. However, given that there was an increased proportion in *high risk* domains, this review demonstrates the need for improvements in the design and implementation of bone tumor RCTs of Ewing's and osteosarcoma.

It is also important to note the significant methodologic challenges associated with the design and implementation of trials in rare diseases such as OS and ES. Trials involving patients with rare tumors face recruitment challenges and long follow-up periods. While methodological bias is present in the literature, it should be noted that trials have led to significant advances in the treatment of bone and soft tissue sarcomas. Prior to the employment of adjuvant and neoadjuvant therapies and novel limb salvage surgeries, the five-year survival of ES and OS used to be 20–30% [29–32]. After the introduction of these medical and surgical advances, the current five-year survival of ES and OS are 60–70%. Therefore, while we identified areas of potential design improvement, the great strides that have been made in this field are commendable [29–32].

This review is strengthened by the novelty of its findings. To our knowledge, this study is the first to assess and the risk of bias of RCTs in osteosarcoma and ES. Secondly, this review was comprehensive in nature, including a large number of RCTs, increasing the robustness of the findings. There are several limitations that must be considered. A limitation of the risk of bias assessment is that it is unable to distinguish between how the trial was conducted and how it was reported. A trial can be methodologically sound but poorly reported and vice-versa. Although the most widely adopted, the Cochrane risk of bias tool is one of many ways clinicians can critically appraise randomized controlled trials [33]. Finally, a potential limitation of the current review is the utility of the Cochrane risk of bias tool. Although this tool has been utilized widely and validated in common diseases, it may be less applicable in rare diseases.

## 5. Conclusions

Sources of potential bias are prevalent among osteosarcoma and ES RCTs, particular in the domains of randomization, allocation concealment, and blinding. Conflicts of interest/industry sponsors were shown to increase the likelihood of having *high risk* domains. Improvements in reporting and adherence to proper methodology, will reduce the risk of bias and improve the validity of the results of RCTs in osteosarcoma and ES.

**Author Contributions:** Conceptualization, R.K. and J.R.P.; methodology R.K.; software, R.K. and A.M.G.; validation, R.K. and J.R.P.; formal analysis, R.K. and A.M.G.; investigation, R.K. and J.R.P.; data curation R.K., A.G., J.R.P.; writing—original draft preparation, R.K. and A.M.G.; writing—review and editing, R.K., A.M.G., J.R.P., A.G., P.F., J.W., K.T.; visualization, R.K., A.M.G., J.R.P.; supervision, J.R.P., A.G., P.F., J.W., K.T.; project administration, R.K., A.G., J.R.P. All authors have read and agreed to the published version of the manuscript.

**Funding:** This research received no external funding.

**Conflicts of Interest:** The authors declare no conflict of interest.

## Appendix A

**Table A1.** List of included studies.

| | Title | Journal | Year | Reference |
|---|---|---|---|---|
| 1 | Transfer factor versus combination chemotherapy: a preliminary report of a randomized postsurgical adjuvant treatment study in osteogenic sarcoma. | Annals of the New York Academy of Sciences | 1976 | [34] |
| 2 | Irradiation of the lungs as an adjuvant therapy in the treatment of osteosarcoma of the limbs. An E.O.R.T.C. randomized study. | European journal of cancer | 1978 | [35] |
| 3 | Delta-9-tetrahydrocannabinol as an antiemetic in cancer patients receiving high-dose methotrexate. A prospective, randomized evaluation. | Annals of internal medicine | 1979 | [36] |
| 4 | Ewing's sarcoma of the vertebral column. | International journal of radiation oncology, biology, physics | 1981 | [37] |
| 5 | The role of radiation therapy in the management of non-metastatic Ewing's sarcoma of bone. Report of the Intergroup Ewing's Sarcoma Study. | International journal of radiation oncology, biology, physics | 1981 | [38] |
| 6 | Japanese experience with clinical trials of fast neutrons. | International journal of radiation oncology, biology, physics | 1982 | [39] |
| 7 | Toxicity associated with combination chemotherapy for osteosarcoma: a report of the cooperative osteosarcoma study (COSS 80). | Journal of cancer research and clinical oncology | 1983 | [40] |
| 8 | VM-26 and dimethyl triazeno imidazole carboxamide in Ewing's sarcoma. | Australian paediatric journal | 1983 | [41] |
| 9 | Adjuvant chemotherapy in osteosarcoma-effects of cisplatinum, BCD, and fibroblast interferon in sequential combination with HD-MTX and adriamycin. Preliminary results of the COSS 80 study. | Journal of cancer research and clinical oncology | 1983 | [42] |
| 10 | The effect of adjuvant chemotherapy on relapse-free survival in patients with osteosarcoma of the extremity. | The New England journal of medicine | 1986 | [43] |
| 11 | Adriamycin-methotrexate high dose versus adriamycin-methotrexate moderate dose as adjuvant chemotherapy for osteosarcoma of the extremities: a randomized study. | European journal of cancer & clinical oncology | 1986 | [44] |
| 12 | A trial of chemotherapy in patients with osteosarcoma (a report to the Medical Research Council by their Working Party on Bone Sarcoma. | British journal of cancer | 1986 | [45] |
| 13 | A randomized study comparing high-dose methotrexate with moderate-dose methotrexate as components of adjuvant chemotherapy in childhood nonmetastatic osteosarcoma: a report from the Childrens Cancer Study Group. | Medical and pediatric oncology | 1987 | [46] |
| 14 | The relationship of various aspects of surgical management to outcome in childhood nonmetastatic osteosarcoma: a report from the Childrens Cancer Study Group. | Journal of pediatric surgery | 1988 | [47] |

**Table A1.** *Cont.*

| | Title | Journal | Year | Reference |
|---|---|---|---|---|
| 15 | Platinum disposition after intraarterial and intravenous infusion of cisplatin for osteosarcoma. Cooperative Osteosarcoma Study Group COSS. | Cancer chemotherapy and pharmacology | 1989 | [48] |
| 16 | Limb sparing versus amputation in osteosarcoma. Correlation between local control, surgical margins and tumor necrosis: Istituto Rizzoli experience. | Annals of oncology: official journal of the European Society for Medical Oncology | 1992 | [49] |
| 17 | Granulocyte-macrophage-colony stimulating factor for prevention of neutropenia and infections in children and adolescents with solid tumors. Results of a prospective randomized study. | Cancer | 1995 | [50] |
| 18 | Radiation therapy in Ewing's sarcoma: an update of the CESS 86 trial. | International journal of radiation oncology, biology, physics | 1995 | [51] |
| 19 | Intra-arterial versus intravenous cisplatinum (in addition to systemic Adriamycin and high dose methotrexate) in the neoadjuvant treatment of osteosarcoma of the extremities. results of a randomized study. | Journal of chemotherapy (Florence, Italy) | 1996 | [52] |
| 20 | Long-term follow-up and post-relapse survival in patients with non-metastatic osteosarcoma of the extremity treated with neoadjuvant chemotherapy. | Annals of oncology: official journal of the European Society for Medical Oncology | 1997 | [53] |
| 21 | Randomised trial of two regimens of chemotherapy in operable osteosarcoma: a study of the European Osteosarcoma Intergroup. | Lancet (London, England) | 1997 | [54] |
| 22 | A multidisciplinary study investigating radiotherapy in Ewing's sarcoma: end results of POG #8346. Pediatric Oncology Group. | International journal of radiation oncology, biology, physics | 1998 | [55] |
| 23 | The pharmacokinetics and metabolism of ifosfamide during bolus and infusional administration: a randomized cross-over study. | British journal of cancer | 1998 | [56] |
| 24 | Ewing sarcoma of the rib: results of an intergroup study with analysis of outcome by timing of resection. | The Journal of thoracic and cardiovascular surgery | 2000 | [57] |
| 25 | Osteosarcoma in preadolescent patients. | Clinical orthopaedics and related research | 2000 | [58] |
| 26 | Granisetron, tropisetron, and ondansetron in the prevention of acute emesis induced by a combination of cisplatin-Adriamycin and by high-dose ifosfamide delivered in multiple-day continuous infusions. | Supportive care in cancer: official journal of the Multinational Association of Supportive Care in Cancer | 2000 | [59] |
| 27 | The possible cost effectiveness of peripheral blood stem cell mobilization with cyclophosphamide and the late addition of G-CSF. | Journal of Korean medical science | 2000 | [60] |
| 28 | Second malignancies after ewing tumor treatment in 690 patients from a cooperative German/Austrian/Dutch study. | Annals of oncology: official journal of the European Society for Medical Oncology | 2001 | [61] |
| 29 | A comparison of methods of loco-regional chemotherapy combined with systemic chemotherapy as neo-adjuvant treatment of osteosarcoma of the extremity. | European journal of surgical oncology: the journal of the European Society of Surgical Oncology and the British Association of Surgical Oncology | 2001 | [62] |

**Table A1.** *Cont.*

| | Title | Journal | Year | Reference |
|---|---|---|---|---|
| 30 | Presurgical chemotherapy compared with immediate surgery and adjuvant chemotherapy for nonmetastatic osteosarcoma: Pediatric Oncology Group Study POG-8651. | Journal of clinical oncology: official journal of the American Society of Clinical Oncology | 2003 | [63] |
| 31 | Addition of ifosfamide and etoposide to standard chemotherapy for Ewing's sarcoma and primitive neuroectodermal tumor of bone. | The New England journal of medicine | 2003 | [64] |
| 32 | Twenty-year follow-up of osteosarcoma of the extremity treated with adjuvant chemotherapy. | Journal of chemotherapy (Florence, Italy) | 2004 | [65] |
| 33 | Treatment of metastatic Ewing's sarcoma or primitive neuroectodermal tumor of bone: evaluation of combination ifosfamide and etoposide—a Children's Cancer Group and Pediatric Oncology Group study. | Journal of clinical oncology: official journal of the American Society of Clinical Oncology | 2004 | [66] |
| 34 | Extracorporeal focused ultrasound surgery for treatment of human solid carcinomas: early Chinese clinical experience. | Ultrasound in medicine & biology | 2004 | [67] |
| 35 | Osteosarcoma: a randomized, prospective trial of the addition of ifosfamide and/or muramyl tripeptide to cisplatin, doxorubicin, and high-dose methotrexate. | Journal of clinical oncology: official journal of the American Society of Clinical Oncology | 2005 | [68] |
| 36 | Ifosfamide, carboplatin, and etoposide (ICE) reinduction chemotherapy in a large cohort of children and adolescents with recurrent/refractory sarcoma: the Children's Cancer Group (CCG) experience. | Pediatric blood & cancer | 2005 | [69] |
| 37 | Local control in pelvic Ewing sarcoma: analysis from INT-0091—a report from the Children's Oncology Group. | Journal of clinical oncology: official journal of the American Society of Clinical Oncology | 2006 | [70] |
| 38 | Contribution to the treatment of nausea and emesis induced by chemotherapy in children and adolescents with osteosarcoma. | Sao Paulo medical journal = Revista paulista de medicina | 2006 | [71] |
| 39 | Intensive therapy with growth factor support for patients with Ewing tumor metastatic at diagnosis: Pediatric Oncology Group/Children's Cancer Group Phase II Study 9457—a report from the Children's Oncology Group. | Journal of clinical oncology: official journal of the American Society of Clinical Oncology | 2006 | [72] |
| 40 | Pharmacokinetics and pharmacodynamics of intravenous epoetin alfa in children with cancer. | Pediatric blood & cancer | 2006 | [73] |
| 41 | Toxicity prevention with amifostine in pediatric osteosarcoma patients treated with cisplatin and doxorubicin. | Pediatric hematology and oncology | 2007 | [74] |
| 42 | SFOP OS94: a randomised trial comparing preoperative high-dose methotrexate plus doxorubicin to high-dose methotrexate plus etoposide and ifosfamide in osteosarcoma patients. | European journal of cancer (Oxford, England: 1990) | 2007 | [75] |
| 43 | Dexrazoxane-associated risk for acute myeloid leukemia/myelodysplastic syndrome and other secondary malignancies in pediatric Hodgkin's disease. | Journal of clinical oncology: official journal of the American Society of Clinical Oncology | | [76] |
| 44 | Improvement in histologic response but not survival in osteosarcoma patients treated with intensified chemotherapy: a randomized phase III trial of the European Osteosarcoma Intergroup. | Journal of the National Cancer Institute | 2007 | [77] |

**Table A1.** *Cont.*

| | Title | Journal | Year | Reference |
|---|---|---|---|---|
| 45 | Therapy-related myelodysplasia and acute myeloid leukemia after Ewing sarcoma and primitive neuroectodermal tumor of bone: A report from the Children's Oncology Group. | Blood | 2007 | [78] |
| 46 | Results of the EICESS-92 Study: two randomized trials of Ewing's sarcoma treatment-cyclophosphamide compared with ifosfamide in standard-risk patients and assessment of benefit of etoposide added to standard treatment in high-risk patients. | Journal of clinical oncology: official journal of the American Society of Clinical Oncology | 2008 | [79] |
| 47 | Osteosarcoma: the addition of muramyl tripeptide to chemotherapy improves overall survival—a report from the Children's Oncology Group. | Journal of clinical oncology: official journal of the American Society of Clinical Oncology | 2008 | [80] |
| 48 | Addition of muramyl tripeptide to chemotherapy for patients with newly diagnosed metastatic osteosarcoma: a report from the Children's Oncology Group. | Cancer | 2009 | [81] |
| 49 | Dose-intensified compared with standard chemotherapy for nonmetastatic Ewing sarcoma family of tumors: a Children's Oncology Group Study. | Journal of clinical oncology: official journal of the American Society of Clinical Oncology | 2009 | [82] |
| 50 | Phase I trial of cixutumumab combined with temsirolimus in patients with advanced cancer. | Clinical cancer research: an official journal of the American Association for Cancer Research | 2011 | [83] |
| 51 | Randomized controlled trial of interval-compressed chemotherapy for the treatment of localized Ewing sarcoma: a report from the Children's Oncology Group. | Journal of clinical oncology: official journal of the American Society of Clinical Oncology | 2012 | [84] |
| 52 | Long-term results (>25 years) of a randomized, prospective clinical trial evaluating chemotherapy in patients with high-grade, operable osteosarcoma. | Cancer | 2012 | [85] |
| 53 | Neoadjuvant chemotherapy with methotrexate, cisplatin, and doxorubicin with or without ifosfamide in nonmetastatic osteosarcoma of the extremity: an Italian sarcoma group trial ISG/OS-1. | Journal of clinical oncology: official journal of the American Society of Clinical Oncology | 2012 | [86] |
| 54 | Cyclophosphamide compared with ifosfamide in consolidation treatment of standard-risk Ewing sarcoma: results of the randomized noninferiority Euro-EWING99-R1 trial. | Journal of clinical oncology: official journal of the American Society of Clinical Oncology | 2014 | [87] |
| 55 | Does intensity of surveillance affect survival after surgery for sarcomas? Results of a randomized noninferiority trial. | Clinical orthopaedics and related research | 2014 | [88] |
| 56 | Methotrexate, Doxorubicin, and Cisplatin (MAP) Plus Maintenance Pegylated Interferon Alfa-2b Versus MAP Alone in Patients with Resectable High-Grade Osteosarcoma and Good Histologic Response to Preoperative MAP: First Results of the EURAMOS-1 Good Response Randomized Controlled Trial. | Journal of clinical oncology: official journal of the American Society of Clinical Oncology | 2015 | [89] |

**Table A1.** *Cont*.

| | Title | Journal | Year | Reference |
|---|---|---|---|---|
| 57 | Impact of gender on efficacy and acute toxicity of alkylating agent -based chemotherapy in Ewing sarcoma: secondary analysis of the Euro-Ewing99-R1 trial. | European journal of cancer (Oxford, England: 1990) | 2015 | [90] |
| 58 | Local control in Ewing sarcoma of the chest wall: results of the EURO-EWING 99 trial. | Annals of surgical oncology | 2015 | [91] |
| 59 | Comparison of MAPIE versus MAP in patients with a poor response to preoperative chemotherapy for newly diagnosed high-grade osteosarcoma (EURAMOS-1): an open-label, international, randomised controlled trial. | The Lancet. Oncology | 2016 | [92] |
| 60 | Zoledronate in combination with chemotherapy and surgery to treat osteosarcoma (OS2006): a randomised, multicentre, open-label, phase 3 trial. | The Lancet. Oncology | 2016 | [93] |
| 61 | Glucagon Decreases IGF-1 Bioactivity in Humans, Independently of Insulin, by Modulating Its Binding Proteins. | The Journal of clinical endocrinology and metabolism | 2017 | [94] |
| 62 | Metronomic Chemotherapy vs. Best Supportive Care in Progressive Pediatric Solid Malignant Tumors: A Randomized Clinical Trial. | JAMA oncology | 2017 | [95] |
| 63 | Results of a randomized, prospective clinical trial evaluating metronomic chemotherapy in nonmetastatic patients with high-grade, operable osteosarcomas of the extremities: A report from the Latin American Group of Osteosarcoma Treatment. | Cancer | 2017 | [96] |
| 64 | The role of FDG PET/CT in patients treated with neoadjuvant chemotherapy for localized bone sarcomas. | European journal of nuclear medicine and molecular imaging | 2017 | [97] |
| 65 | Significance of neoadjuvant chemotherapy (NACT) in limb salvage treatment of osteosarcoma and its effect on GLS1 expression. | European review for medical and pharmacological sciences | 2018 | [98] |
| 66 | Ewing's Sarcoma of the Head and Neck: Margins are not just for surgeons. | Cancer medicine | 2018 | [99] |
| 67 | Comprehensive Treatment and Rehabilitation of Patients with Osteosarcoma of the Mandible. | Implant dentistry | 2018 | [100] |
| 68 | Pantoprazole, an Inhibitor of the Organic Cation Transporter 2, Does Not Ameliorate Cisplatin-Related Ototoxicity or Nephrotoxicity in Children and Adolescents with Newly Diagnosed Osteosarcoma Treated with Methotrexate, Doxorubicin, and Cisplatin. | The Oncologist | 2018 | [101] |
| 69 | Gabapentin as an Adjuvant Therapy for Prevention of Acute Phantom-Limb Pain in Pediatric Patients Undergoing Amputation for Malignant Bone Tumors: A Prospective Double-Blind Randomized Controlled Trial. | Journal of pain and symptom management | 2018 | [102] |
| 70 | Results of methotrexate-etoposide-ifosfamide based regimen (M-EI) in osteosarcoma patients included in the French OS2006/sarcome-09 study. | European journal of cancer (Oxford, England: 1990) | 2018 | [103] |
| 71 | Efficacy and safety of regorafenib in adult patients with metastatic osteosarcoma: a non-comparative, randomised, double-blind, placebo-controlled, phase 2 study. | The Lancet. Oncology | 2019 | [104] |

**Table A1.** *Cont.*

| | Title | Journal | Year | Reference |
|---|---|---|---|---|
| 72 | Effects of mindfulness-based stress reduction combined with music therapy on pain, anxiety, and sleep quality in patients with osteosarcoma. | Revista brasileira de psiquiatria (Sao Paulo, Brazil: 1999) | 2019 | [105] |
| 73 | Addition of Zoledronate to Chemotherapy in Patients with Osteosarcoma Treated with Limb-Sparing Surgery: A Phase III Clinical Trial. | Medical science monitor: international medical journal of experimental and clinical research | 2019 | [106] |

**Table A2.** Overall risk of bias stratified by type of sarcoma (osteosarcoma vs. Ewing sarcoma), type of intervention (medical vs. surgical), and presence or absence of metastasis.

| | Study Stratifica-tion | (N) | Overall Percentages of Risk of Bias in All Domains | | | Domain with the Most *"High-Risk"* Ratings | Domain with the Most *"Unclear-Risk"* Ratings |
|---|---|---|---|---|---|---|---|
| | | | *Low Risk* | *Unclear Risk* | *High-Risk* | | |
| Type of Sarcoma | Osteosarcoma (OA) | 46 | 49.6% (137/276) | 43.1% (119/276) | 7.2% (20/276) | Blinding of participants and personnel | Blinding of participants and personnel |
| | Ewing Sarcoma (ES) | 23 | 46.0% (74/161) | 51.6% (83/161) | 2.5% (4/161) | Blinding of participants and personnel | Blinding of outcome assessment |
| | RCTs containing both OA and ES | 4 | 53.6% (15/28) | 46.2% (13/28) | 0% (0/28) | - | Blinding of participant, personnel, and outcome assessment |
| Type of Intervention | Medical RCTs | 69 | 47.6% (230/483) | 47.4% (229/483) | 5.0% (24/483) | Blinding of participants and personnel | Blinding of outcome assessment |
| | Surgical RCTs | 4 | 42.9% (12/28) | 53.6% (15/28) | 3.6% (1/28) | Random Sequence Generation | Allocation sequence concealment, and blinding of participant, personnel, and outcome assessment |
| Presence/ Absence of Metastasis | RCTs solely on metastatic disease | 5 | 51.4% (18/35) | 48.6% (17/35) | 0% (0/35) | - | Blinding of participant, personnel, and outcome assessment |
| | RCTs solely on non-metastatic disease | 5 | 45.7% (16/35) | 54.3% (19/35) | 0% (0/35) | - | Allocation sequence concealment, and blinding of participant, personnel, and outcome assessment |
| | RCT not stratified based on presence or absence of metastasis | 63 | 47.0% (203/432) | 47.2% (204/432) | 5.8% (25/432) | Blinding of participant, personnel, and outcome assessment | Blinding of participant, personnel, and outcome assessment |

**Table A3.** Studies with any domain with a *high risk* of bias rating.

| Study | Type of Sarcoma | Intervention | Journal | Year |
|---|---|---|---|---|
| Local control in pelvic Ewing sarcoma: analysis from INT-0091—a report from the Children's Oncology Group. | Ewing Sarcoma | Medical | Journal of clinical oncology: official journal of the American Society of Clinical Oncology | 2006 |
| Second malignancies after ewing tumor treatment in 690 patients from a cooperative German/Austrian/Dutch study. | Ewing Sarcoma | Medical | Annals of oncology: official journal of the European Society for Medical Oncology | 2001 |
| Randomised trial of two regimens of chemotherapy in operable osteosarcoma: a study of the European Osteosarcoma Intergroup. | Osteosarcoma | Medical | Lancet (London, England) | 1997 |
| Dexrazoxane-associated risk for acute myeloid leukemia/myelodysplastic syndrome and other secondary malignancies in pediatric Hodgkin's disease. | Osteosarcoma | Medical | Journal of clinical oncology: official journal of the American Society of Clinical Oncology | 2007 |
| Japanese experience with clinical trials of fast neutrons. | Osteosarcoma | Medical | International journal of radiation oncology, biology, physics | 1982 |
| Pharmacokinetics and pharmacodynamics of intravenous epoetin alfa in children with cancer. | Osteosarcoma | Medical | Pediatric blood & cancer | 2006 |
| Limb sparing versus amputation in osteosarcoma. Correlation between local control, surgical margins and tumor necrosis: Istituto Rizzoli experience. | Osteosarcoma | Surgical | Annals of oncology: official journal of the European Society for Medical Oncology | 1992 |
| Cyclophosphamide compared with ifosfamide in consolidation treatment of standard-risk Ewing sarcoma: results of the randomized noninferiority Euro-EWING99-R1 trial. | Ewing Sarcoma | Medical | Journal of clinical oncology: official journal of the American Society of Clinical Oncology | 2014 |
| Contribution to the treatment of nausea and emesis induced by chemotherapy in children and adolescents with osteosarcoma. | Osteosarcoma | Medical | Sao Paulo medical journal = Revista paulista de medicina | 2006 |
| Improvement in histologic response but not survival in osteosarcoma patients treated with intensified chemotherapy: a randomized phase III trial of the European Osteosarcoma Intergroup. | Osteosarcoma | Medical | Journal of the National Cancer Institute | 2007 |
| Methotrexate, Doxorubicin, and Cisplatin (MAP) Plus Maintenance Pegylated Interferon Alfa-2b Versus MAP Alone in Patients with Resectable High-Grade Osteosarcoma and Good Histologic Response to Preoperative MAP: First Results of the EURAMOS-1 Good Response Randomized Controlled Trial. | Osteosarcoma | Medical | Journal of clinical oncology: official journal of the American Society of Clinical Oncology | 2015 |

<div align="center">**Table A3.** *Cont*.</div>

| Study | Type of Sarcoma | Intervention | Journal | Year |
|---|---|---|---|---|
| Comparison of MAPIE versus MAP in patients with a poor response to preoperative chemotherapy for newly diagnosed high-grade osteosarcoma (EURAMOS-1): an open-label, international, randomised controlled trial. | Osteosarcoma | Medical | The Lancet. Oncology | 2016 |
| Effects of mindfulness-based stress reduction combined with music therapy on pain, anxiety, and sleep quality in patients with osteosarcoma. | Osteosarcoma | Medical | Revista brasileira de psiquiatria (Sao Paulo, Brazil: 1999) | 2019 |
| Zoledronate in combination with chemotherapy and surgery to treat osteosarcoma (OS2006): a randomised, multicentre, open-label, phase 3 trial. | Osteosarcoma | Medical | The Lancet. Oncology | 2016 |
| Does intensity of surveillance affect survival after surgery for sarcomas? Results of a randomized noninferiority trial. | Osteosarcoma | Medical | Clinical orthopaedics and related research | 2014 |

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
