# Peer review of "Assessment of Risk of Bias in Osteosarcoma and Ewing’s Sarcoma Randomized Controlled Trials: A Systematic Review"

_curroncol, doi:10.3390/curroncol28050322_

Round 1

Reviewer 1 Report

Dear Editor I read the article Assessment of Risk of Bias in Osteosarcoma and Ewing’s Sarcoma Randomized Controlled Trials: A Systematic Review  and I have some concerns.

Everybody who studies and treats patients with rare tumors (  tumours incidence  <5 cases /100 000/years) knows the great  difficulty in carrying out a study on this subject.

Ewing sarcomas (ES) and Osteosarcomas (OS)are prototype of rare tumors :

ES accounts for 0,5 cases/100000/years and OS for 1,2 cases /100 000 / years.Moreover we have to distinguish paediatric ES and OS and adult bone sarcomas. This is to explain  the effort to plan , conduct, conclude and publish a study on these neoplasms.

We all know that bias are present in the studies on rare tumors and not only for methodological and statistical reasons: different site of primary tumors ( spine and pelvis versus extremities), long time  of enrollment ( median 6 years for a neoadjuvant / adjuvant study), the different  quality of surgery, the level of necrosis obtained after therapy, the age of the patients and so on.

We totally agree with the Authors  that rare tumors studies can show   risk of bias as  blinding of participants and personnel, blinding of outcome assessors,  randomization and allocation concealment  but it is universally  recognized that important improvements have been obtained in ES and OS since 1970's.

Remember that before adjuvant and neoadjuvat studies ES 5 years survival was 20% , now is 60%; OS 5 years survival was 30% now is 70%. In spite of the methodological bias  great advances were recorded. The Authors should point out these extraordinary results in the discussion.

Moreover the Authors  cite 73 studies included  for the analysis. The titles of the articles should be reported in order to allow the reader's evaluation.

In a summary table the articles should be divided on the basis of tumor ( ES or OS) , local disease versus metastatic disease, study on surgery and on medical treatment, in order to recognise the risk of bias stratified for group of risk.

In conclusion this article has a high methodological profile, but the negative aspects  cannot forget the important improvement in clinical oncology recorded in ES and OS .

Reviewer 2 Report

Dear Authors, 

The study represents a systematic review to determine the prevalence of risk of bias in primary bone cancer RCTs, impact of conflict of interest and industry and sponsorship on risk of bias as well changes in quality of RCTs over time. 

The manuscript is very well written, the methodology and conclusions are very valid. 

A minor suggestion, if was feasible to add a subgrouping analysis that assess the same objective parameters of Ewing and osteosarcoma separately?

Thank you,

Reviewer 3 Report

first of all I would congratulate the authors with this paper, because I think this covers an highly important issue in the process of interpretation of clinical trials, particularly RCTs. Because of the lack of progress in clinical improvement in treatment of the 2 primary bone tumors, clinical investigators need to be aware what factors can contribute to this failure. One of these fcators is that we lean on RCTs, which are difficult to perform in rare diseases as OS and EWS. So the importance of critical looking at these studies is without any doubt. 

However, I really miss details about the reported bias, as suggested by the paper, in the OS and EWS trial. I would like to have more insight in the bias as found, particularly of the trials with bias of high risk rating. I would suggest the authors can provide these details in an addendum file.

Secondly, trial in the early years of treatment (prior to 1996 fofr example) need to be approached in the analysis in another way than trials, performed after the year 2000. In the discussion this is not recognized as such. So I would ask thte authors if they could make a useful remark about the time-effect on the reporting of bias. 

Last remark is about blinding, which in my view is impossible in trials with rare diseases, such as these bone tumors. I'm curious to know in how far the authors think blinding as bias will really influence the outcome data compared to  not blinded RCT's, and as such will influence clinical practice. 

Round 2

Reviewer 1 Report

Looking at the first version I realize that some changes were done in the last part of the Discussion and in the related tables.

However there are some  points that have not been fully considered:

1) Osteosarcoma and Ewing sarcoma are rare tumors and RCT are difficult to be planned and  conducted for the low number of Patients and the long time of enrollment. Probably the most important bias should be identify in these situations and not only in the conflict of interest of the Authors

2) The Cochrane cooperation analysis is well   applicable in the frequent tumors, but  are  not fully  adaptable  in uncommon tumor settings

3) Many statisticians  propose to apply in rare tumor a Bayesian  analysis and not the classical null hypothesis ( p<0,05) or Kaplan Meyer analysis.

Round 3

Reviewer 1 Report

Some changes have been done and now the article clearly  expresses  the problems of the researches on rare tumors.

Many problems come from the rarity of the neoplasms and the weakness of the studies   are related to the different approach in diagnosis and therapy of ES and OS.

Bias from RCT are partially due   to the sponsorhip of the studies but many other problems come from the lack of innovative therapies and drugs.

I congratulate the Authors  who accepted the suggestions and reviewed the paper